# Dual-comb spectroscopic ellipsometry

Takeo Minamikawa [1,2], Yi-Da Hsieh[1,2], Kyuki Shibuya[1,2], Eiji Hase[1,2], Yoshiki Kaneoka[1], Sho Okubo[2,3], Hajime Inaba[2,3], Yasuhiro Mizutani[2,4], Hirotsugu Yamamoto[2,5], Tetsuo Iwata[1,2] & Takeshi Yasui[1,2]

Spectroscopic ellipsometry is a means of investigating optical and dielectric material responses. Conventional spectroscopic ellipsometry is subject to trade-offs between spectral accuracy, resolution, and measurement time. Polarization modulation has afforded poor performance because of its sensitivity to mechanical vibrational noise, thermal instability, and polarization-wavelength dependency. We combine spectroscopic ellipsometry with dual-comb spectroscopy, namely, dual-comb spectroscopic ellipsometry. Dual-comb spectroscopic ellipsometry (DCSE). DCSE directly and simultaneously obtains the ellipsometric parameters of the amplitude ratio and phase difference between s-polarized and p-polarized light signals with ultra-high spectral resolution and no polarization modulation, beyond the conventional limit. Ellipsometric evaluation without polarization modulation also enhances the stability and robustness of the system. In this study, we construct a polarization-modulation-free DCSE system with a spectral resolution of up to $1.2 \times 10^{-5}$ nm throughout the spectral range of 1514–1595 nm and achieved an accuracy of 38.4 nm and a precision of 3.3 nm in the measurement of thin-film samples.

[1] Graduate School of Technology, Industrial and Social Sciences, Tokushima University, 2-1 Minami-Josanjima, Tokushima 770-8506, Japan. [2] Japan Science and Technology Agency (JST), ERATO Intelligent Optical Synthesizer (IOS) Project, 2-1 Minami-Josanjima, Tokushima 770-8506, Japan. [3] National Metrology Institute of Japan (NMIJ), National Institute of Advanced Industrial Science and Technology (AIST), 1-1-1 Umezono, Tsukuba, Ibaraki 305-8563, Japan. [4] Graduate School of Engineering, Osaka University, 2-1 Yamadaoka, Suita, Osaka 565-0871, Japan. [5] Center for Optical Research and Education, Utsunomiya University, 7-1-2 Yoto, Utsunomiya, Tochigi 321-8585, Japan. Takeo Minamikawa and Yi-Da Hsieh contributed equally to this work. Correspondence and requests for materials should be addressed to T.M. (email: minamikawa.takeo@tokushima-u.ac.jp)

Spectroscopic ellipsometry (SE) is widely used for the investigation and evaluation of materials in terms of optical and dielectric response in both academic and industrial research. The applications of ellipsometry include the optical characterization of thin films[1–3], the in situ monitoring of semiconductor processing[4, 5], surface sensing in biochemical reactions[6–8], and the evaluation of graphene[9–11]. SE measures the polarization state of light incident upon a material as a function of wavelength to determine the properties of the material, such as the complex dielectric function, carrier structure, crystalline nature, and thickness of a thin film[12–14]. The complex reflectance ratio $\rho$ for ellipsometric analysis is given by:

$$\rho = \tan(\Psi)\exp(i\Delta), \qquad (1)$$

where $\Psi$ and $\Delta$ denote the amplitude ratio and phase difference, respectively, between the p-polarization and s-polarization components of the polarization state of the incident light, or the so-called ellipsometric parameters. These ellipsometric parameters are obtained as a function of wavelength. In conventional SE, the ellipsometric parameters are obtained from measurements of the optical intensity resulting from the modulation of the polarization state of the light interacting with the target material. In general, a multi-channel dispersive spectrometer equipped with a rotating polarizer is used to obtain ellipsometric parameters with a wide spectral bandwidth. However, this mechanical polarization modulation limits the mechanical stability of conventional SE. Although a photoelastic modulator (PEM) or electro-optic modulator (EOM) can be used for non-mechanical polarization or phase modulation, the wavelength and/or temperature dependency of the polarization modulation provided by such devices is a limiting factor. Furthermore, the fast polarization modulation imposed by a PEM or EOM (typically, several tens of kilohertz to a few tens of megahertz) makes it difficult to combine with the use of a multi-channel spectrometer equipped with a CCD or CMOS camera. Therefore, such conventional non-mechanical polarization modulation is not compatible with real-time SE. In addition, high spectral resolution is important in the analysis of highly wavelength-dependent samples and thick-film samples. Although the spectral resolution of conventional SE is typically $10^{-1}$ to $10^{-2}$ nm (1–10 GHz) with the use of dispersive spectrometers, this resolution is sometimes insufficient for such samples.

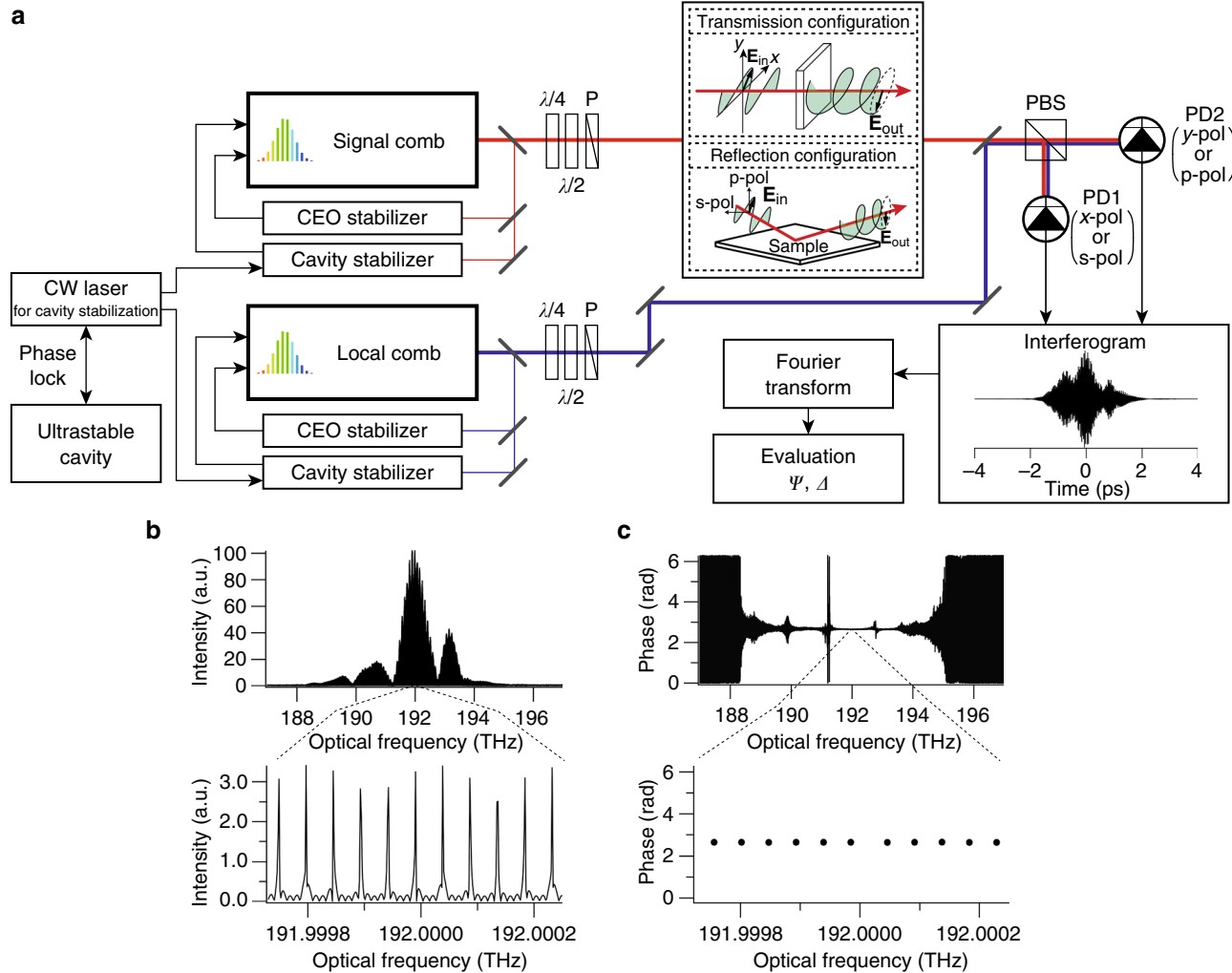

**Fig. 1** Experimental setup and fundamental specifications of the developed dual-comb spectroscopic ellipsometry (DCSE) system. **a** Configuration of the laser stabilization system and optical setup of the DCSE system. The components include a continuous wave (CW) laser locked to an ultra-stable cavity, a quarter-wave plate ($\lambda$/4), a half-wave plate ($\lambda$/2), a polarizer (P), a polarization beam splitter (PBS), and a photodetector (PD); the possible polarization components include p-polarization (p-pol), s-polarization (s-pol), x-polarization (x-pol), and y-polarization (y-pol) components. Fundamental specifications in the form of **b** an amplitude spectrum of p-polarized light and **c** a phase difference spectrum of p-polarized and s-polarized light. The comb modes are separated by the repetition frequency of 48 MHz ($3.8 \times 10^{-4}$ nm) and can be resolved down to 1.5 MHz ($1.2 \times 10^{-5}$ nm)

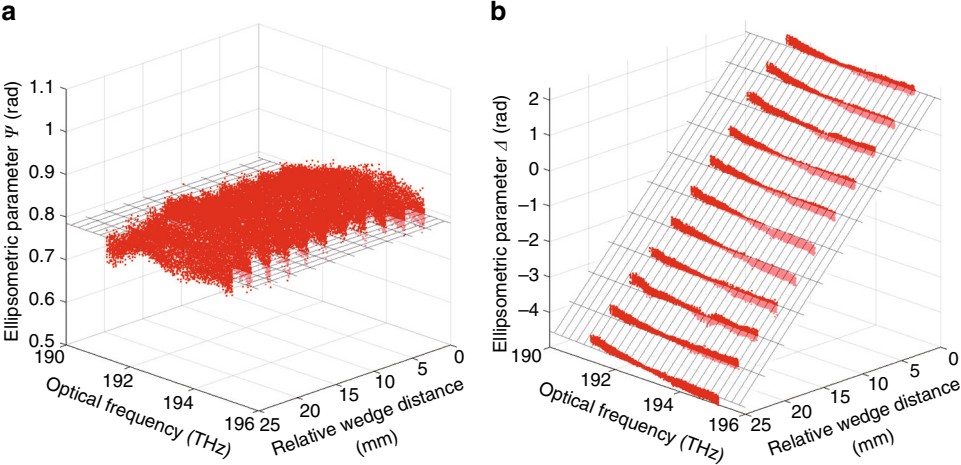

**Fig. 2** Ellipsometric evaluation of a Soleil-Babinet compensator using the DCSE system. The ellipsometric parameters, **a** $\Psi$ and **b** $\Delta$, obtained using the DCSE system (*red dots*) and through theoretical estimation (*mesh surface*)

To overcome these limitations, we propose SE employing dual-comb spectroscopy, namely, dual-comb spectroscopic ellipsometry (DCSE). Dual-comb spectroscopy is a promising technique for ultra-high resolution, high accuracy, broadband spectroscopy with rapid data acquisition[15–30]. Dual-comb spectroscopy employs two optical comb lasers with slightly different repetition rates, allowing the acquisition of a complete interferogram with an ultra-wide time span without mechanical scanning for the deduction of a highly resolved optical spectrum[29]. Since an optical comb laser is quite stable because of the stabilization of the repetition rate and carrier envelope offset, the frequency of each comb mode can be accurately and precisely determined to within $10^{-6}$ to $10^{-4}$ nm (100 kHz–10 MHz)[29, 31]. Furthermore, since dual-comb spectroscopy is based on Fourier transform spectroscopy, the full amplitude and phase spectra can be obtained through directly decoding from the interferograms of two optical-comb lasers. However, in previous research on gas spectroscopy, the use of only an amplitude spectrum has been found to be sufficient for the evaluation of fine spectral structures[29]. In our proposed method, we utilize both the amplitude and phase spectra along two orthogonal axes, enabling precise polarization analysis without the need for mechanical or electro-optic polarization modulation. These advantages of dual-comb spectroscopy enable SE with ultra-high spectral resolution and accuracy in a moderate spectral range for deducing spectra and polarization states without any polarization modulation.

In this study, we provide a proof-of-principle demonstration of DCSE for novel material evaluation. We developed a DCSE system to demonstrate the ability to perform ellipsometric analysis without any polarization modulation by using two optical comb laser sources employing erbium-doped-fiber-based mode-locked lasers. DCSE evaluations of birefringent materials and thin films are demonstrated using the reflection and transmission configurations, respectively.

## Results

**Principle of operation**. In DCSE, the ellipsometric parameters are evaluated in terms of the Fourier transformation of interferograms observed in the p-polarization and s-polarization components in a reflection configuration or the x-polarization and y-polarization components in a transmission configuration (Fig. 1a; see also "Methods"). Two highly stabilized, amplified comb lasers with repetition rates of $f_{rep,S}$ and $f_{rep,L}$

were synchronized with slightly different repetition rates ($\Delta f_{rep} = f_{rep,S} - f_{rep,L}$). The temporal interferogram of the two comb lasers had a sampling interval of $\Delta t = 1/f_{rep,L} - 1/f_{rep,S}$. The amplitude and phase of each polarization component were decoded from the interferograms by means of Fourier transform spectroscopy. When the light from a comb laser (the signal comb laser) interacts with a sample, the resultant change in the polarization of the light is directly encoded, with both amplitude and phase, into the interferograms.

The ellipsometric parameters $\Psi$ and $\Delta$ were thus evaluated by decoding the interferograms obtained from the photodetectors (PDs) as follows:

$$\Delta = \arg\left(\mathbf{I}_{PD1}\right) - \arg\left(\mathbf{I}_{PD2}\right) + \pi, \qquad (2)$$

$$\Psi = \tan^{-1}\frac{\left|\mathbf{I}_{PD1}\right|}{\left|\mathbf{I}_{PD2}\right|}, \qquad (3)$$

where $\mathbf{I}_{PD1}$ and $\mathbf{I}_{PD2}$ denote the individually observed signals of each polarization recorded by PD1 and PD2, of which the amplitude ($|\mathbf{I}|$) and phase ($\arg(\mathbf{I})$) can be decoded via Fourier transformation. Since the angles of the polarizers at the signal comb and local comb arms were set to $\pi/4$ and $-\pi/4$, respectively, in our setup, the phase difference of each arm initially had a phase shift of $\pi$, as shown in Eq. (2). The evaluation of $\Psi$ and $\Delta$ does not require mechanical or electro-optic modulation of the polarization states, indicating that polarization-modulation-free SE with a moderate spectral bandwidth can be realized.

**Basic spectral performance of DCSE**. The basic spectral performance of the DCSE system was evaluated using a gold mirror in the reflection configuration, with the results illustrated in Fig. 1b, c. Here, we define the spectral resolution as the linewidth of the comb modes[32]. This is because the ultimate resolution of dual-comb spectroscopy can reach the linewidth, whereas the spectral sampling points are discrete, with a spectral spacing that coincides with the repetition rate of the comb sources. By applying an interferogram concatenation method[27, 33] with 20 duplications of a single interferogram with one complete period of the repetition rate ($1/\Delta f_{rep}$), we achieved an ultra-high spectral resolution of 1.5 MHz ($1.2 \times 10^{-5}$ nm) and a spectral spacing of 48 MHz ($3.8 \times 10^{-4}$ nm) in the spectral range of 1514–1595 nm (188–198 THz), as shown in the amplitude spectrum presented in Fig. 1b. Since $\Psi$ was examined using the

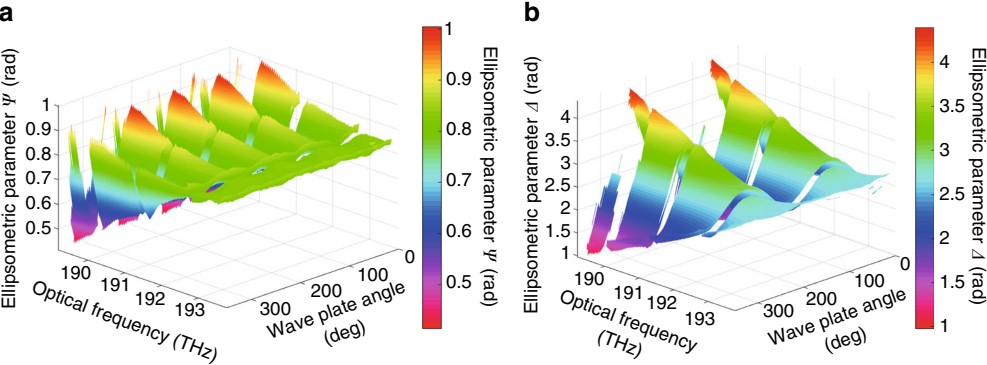

**Fig. 3** Ellipsometric evaluation of a high-order wave plate using the DCSE system. The ellipsometric parameters, **a** Ψ and **b** Δ, obtained using the DCSE system at a variety of rotation angles of the high-order wave plate

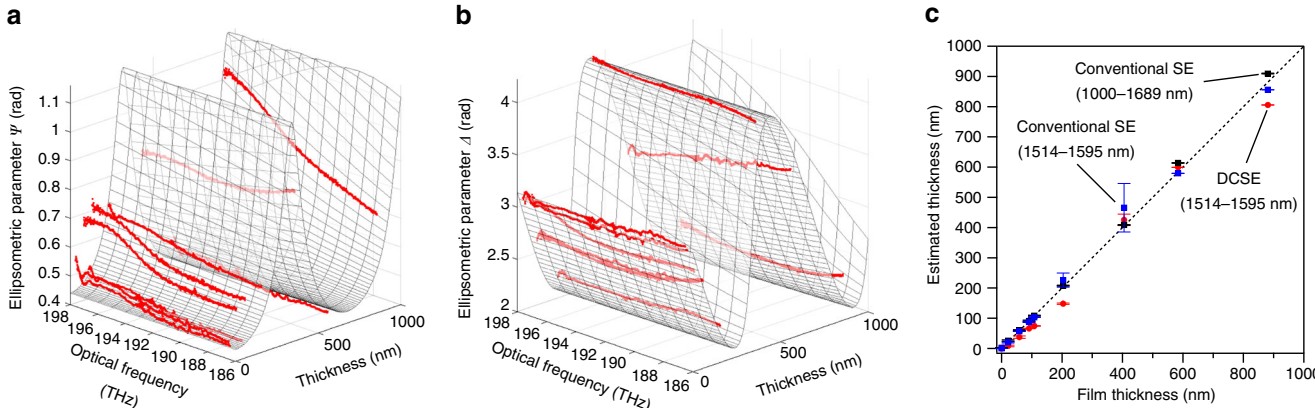

**Fig. 4** Ellipsometric evaluations of thin-film samples using the DCSE system. The ellipsometric parameters, **a** Ψ and **b** Δ, obtained using the DCSE system (*red dots*) and through theoretical estimation (*mesh surfaces*). **c** Thicknesses determined via DCSE with the spectral range of 1514–1595 nm (*red*) and via conventional SE with the spectral range of 1000–1689 nm (*black*) or 1514–1595 nm (*blue*). *Error bars indicate the standard deviation.*

p-polarization and s-polarization components of each comb mode, the spectral resolution of Ψ coincided with that of the amplitude spectra of the p-polarization and s-polarization components. The phase spectrum was constructed by extracting the phase data at the optical frequency of each comb mode peak of the amplitude spectrum. The spectral resolution of the phase spectrum also coincided with that of the amplitude spectrum, as shown in Fig. 1c. Furthermore, the spectral interleaving method[27, 29] or multi-interferogram observation[34], which is generally used in dual-comb spectroscopy, can be used in DCSE to achieve a higher spectral resolution.

**Soleil-Babinet compensator**. As a first experimental demonstration, we performed an ellipsometric evaluation of a Soleil-Babinet compensator in the transmission configuration. The ellipsometric parameters of the Soleil-Babinet compensator observed using the DCSE system with a variety of relative wedge distances are shown as *red dots* in Fig. 2 and in Supplementary Movies 1, 2. With the signal-to-noise ratio (SNR) defined as the peak spectral amplitude divided by the standard deviation of the noise spectral region, the typical SNR was 235. The theoretical relationships between the ellipsometric parameters and the relative wedge distance, which were calibrated at an optical frequency of 1550 nm (193.4 THz), are also shown as *mesh surfaces* in Fig. 2. Since the fast axis of the Soleil-Babinet compensator was aligned along the axis of x-polarization in the sample coordinate system, the Soleil-Babinet compensator served

as a relative phase retarder of the x-polarization and y-polarization components of the signal comb laser without causing any change in amplitude relative to each polarization. The ellipsometric parameters, Ψ and Δ, determined using the DCSE system reflected the following optical conditions. The flat structure of Ψ indicated that the relative amplitudes of the x-polarization and y-polarization components were insensitive to the relative wedge distance of the Soleil-Babinet compensator and the optical frequency. In addition, the linear relationship between Δ and the relative wedge distance of the Soleil-Babinet compensator was that of a phase retarder relative to the x-polarization and y-polarization components. The DCSE estimation errors for Ψ and Δ are shown in Supplementary Fig. 1. We obtained root mean square errors (RMSEs) of 19.3 and 58.0 mrad for Ψ and Δ;, respectively, at the calibrated optical frequency of 1550 nm (193.4 THz). We obtained RMSEs of 24.5 and 109 mrad for Ψ and Δ, respectively, for the optical frequency range of 1537–1570 nm (191–195 THz). These results clearly support the assertion that DCSE has the potential to be used for polarization analysis along two orthogonal axes for the ellipsometric analysis of birefringent materials.

**High-order wave plates**. To evaluate the detection capability for an optical-frequency-dependent birefringent material, we performed an ellipsometric evaluation of a high-order wave plate in the transmission configuration. Figure 3 and Supplementary Movies 3, 4 show the DCSE evaluation of a high-order wave plate

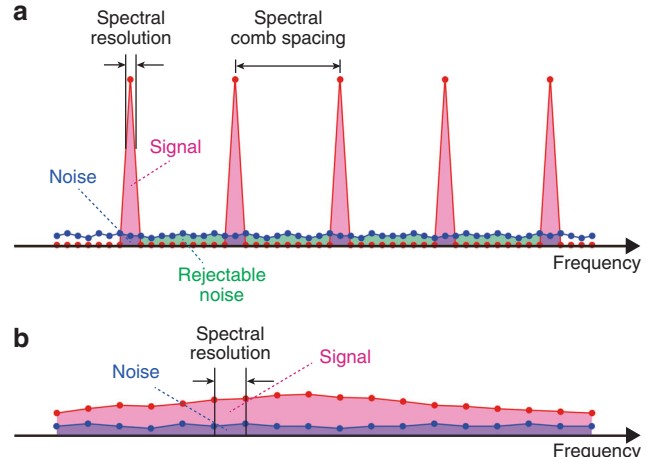

**Fig. 5** Signal-to-noise ratio (SNR) enhancement in ellipsometric measurements with high spectral resolution. **a** The noise components existing in the frequency gaps between comb modes can be rejected in a mode-resolved optical comb spectrum. **b** No noise rejection is possible with low spectral resolution of the continuous-spectrum light. The *red*-shaded, *blue*-shaded, and *green*-shaded areas represent the signal components, the noise components that contribute to the SNR of ellipsometric measurements, and the noise components that can be rejected and is not contributed to the SNR in ellipsometric evaluation

designed as a 30th-order quarter-wave plate at 633 nm, which also functions as a high-order wave plate in the 1550 nm region. A typical SNR of 800 was obtained. Throughout the 360° rotation of the high-order wave plate with respect to the linear polarization of the incident light, we obtained four peaks in $\Psi$ and two peaks in $\Delta$. These findings are in good agreement with the incidence-angle-dependent $\Psi$ and $\Delta$ behavior of the optically uniaxial phase retardation of a general wave plate. Regarding the optical frequency, we observed a rotation-angle-insensitive region at 1549 nm (193.5 THz), which indicated that the phase retardation between the $x$-polarization and $y$-polarization components was equal to an integer multiple of $2\pi$. This result supports the application of DCSE in ellipsometric evaluations of optical-frequency-dependent birefringent materials.

**SiO$_2$ thin-film samples**. Finally, we applied the proposed DCSE system for the ellipsometric evaluation of thin films in the reflection configuration. We determined the thicknesses of thin-film samples consisting of SiO$_2$ thin films 0–900 nm in thickness deposited on a silicon base plate. Figure 4a, b and Supplementary Movies 5, 6 show that $\Psi$ and $\Delta$ depend on the film thickness at an incidence angle of 58°. A typical SNR of 246 was obtained. The experimental results for $\Psi$ and $\Delta$ (*red dots* in Fig. 4a, b) were in good agreement with the theoretical estimates (*mesh surfaces* in Fig. 4a, b) calculated using a three-layer model with a structure of air/SiO$_2$/Si.

We measured the film thickness by minimizing the RMSE of the experimental and theoretical values of $\Delta$ over a spectral range of 1514–1595 nm (188–198 THz). A local minimum in the RMSE was observed approximately every 650 nm because of the insufficient spectral bandwidth used for the film thickness determination (Supplementary Fig. 2). We determined the film thickness by evaluating the local minimum within 500 nm of the theoretical estimate. The film thicknesses thus determined are shown as *red dots* in Fig. 4c. The RMSE and precision of the repeated observation of the film thickness results were 38.4 and 3.3 nm (see Fig. 4c and Supplementary Fig. 3), which correspond to the accuracy and precision, respectively, of the film thickness

determination. For comparison, we performed the same experiment using a commercially available SE system with mechanical polarization rotation, namely, conventional SE (M-2000DI-YK, J. A. Woollam, spectral range of 1000–1689 nm, spectral resolution of 3.4 nm coinciding with the spectral spacing, number of data points of approximately 200), with the results shown as *black points* in Fig. 4c, and achieved an RMSE of 10.0 nm and a thickness precision of 1.2 nm (see Fig. 4c and Supplementary Fig. 3). These values were somewhat better than those achieved using the DCSE system (spectral range of 1514–1595 nm, spectral resolution of $1.2 \times 10^{-5}$ nm, spectral spacing of $3.8 \times 10^{-4}$ nm, number of data points of approximately 200,000). However, it is important to note that there is a large difference in spectral bandwidth between the two systems considered in this comparison and that the fitting accuracy for film thickness determination largely depends on the spectral range. In other words, DCSE has a potential to achieve moderate ellipsometric performance for a much narrower spectral bandwidth than that used in conventional SE. The present performance achieved in such a limited spectral range may benefit from the high spectral resolution of DCSE and/or the consequently higher number of spectral data points, as well as the higher stability by virtue of the lack of mechanical polarization modulation. Since state-of-the art dual-comb sources have a spectral bandwidth comparable to that available in conventional SE[28], the use of such broadband comb sources in DCSE is a promising way to enhance the performance to a level equal to or greater than that of conventional SE.

To compare DCSE and conventional SE in the same spectral range, we reduced the spectral range of conventional SE to be equal to that of DCSE (spectral range of 1514–1595 nm, number of data points of 27), and then performed the same experiment as indicated *blue points* in Fig. 4c and Supplementary Fig. 3. The resulting RMSE and thickness precision were 28.9 and 12.1 nm, respectively. Therefore, RMSE achieved using DCSE is almost comparable to that using conventional SE, whereas the thickness precision using DCSE was significantly better than that using conventional SE. A little difference of RMSE between them is mainly due to the fact that the instrumental calibration was not performed using the film thickness standard in DCSE. More precise calibration of the optical setup of the DCSE system (incidence angle, systematic phase error, and so on) would reduce the systematic error, and hence improve its performance.

## Discussion

We first discuss the importance of ultra-high spectral resolution in SE. High spectral resolution is generally required in the ellipsometric evaluation of highly wavelength-dependent materials, as shown in Fig. 3. Further applications are expected in the evaluation of absorbable materials with vibrational transitions, for instance, in the mid-infrared or far-infrared region[35–37]. In such ellipsometric applications, the high spectral resolution of DCSE can be effectively utilized for the precise and accurate evaluation of material properties. Even if the theoretical models of the dielectric functions are not well known, DCSE can at least be used to evaluate the spectral dependency of the ellipsometric parameters $\Psi$ and $\Delta$. High spectral resolution characterization of this spectral dependency might provide helpful data for the estimation of theoretical models.

Ultra-high spectral resolution is also beneficial for determining the thicknesses of film samples without any noticeable wavelength dependence of the refractive index and absorption behavior. In general, the spectral resolution is chosen to avoid under-sampling of the fringe spacing in the spectral interference. However, the fringe spacing narrows in proportion to an increase in the film thickness, especially in the short-wavelength region with normal

dispersion. Therefore, the maximum measurable thickness in conventional SE is limited by its spectral resolution. The ultra-high spectral resolution of DCSE throughout the broad visible region has the potential to greatly enhance the maximum measurable film thickness.

More interestingly, the ultra-high spectral resolution of DCSE also offers the benefit of improving the SNR of ellipsometric measurements. An optical frequency comb has a discrete, localized distribution of optical energy with a constant frequency spacing in the optical frequency region, and dual-comb spectroscopy enables us to acquire the entire range of optical energy in the radio-frequency region without loss, as shown in Fig. 5a. By contrast, the noise component is not localized but rather is continuously distributed throughout the entire radio-frequency region. The ultra-high spectral resolution of DCSE in acquiring the mode-resolved optical comb spectrum enables us to select only the signal and noise components around the comb modes and to reject the remaining noise components existing in the frequency gaps between comb modes. Although the noise rejection efficiency depends on the ratio of the spectral comb spacing to the comb mode linewidth (typically, >10), the SNR should be greatly enhanced at the positions of the comb modes. By contrast, in conventional SE using a broadband continuous-spectrum light source, since both the signal and noise components are not localized and are continuously distributed, as shown in Fig. 5b, all noise components contribute to the SNR; that is, there is no noise rejection effect. If the total energy of the optical and noise components in DCSE (total area of the signal and noise components in Fig. 5a) is equal to that in conventional SE (total area of the signal and noise components in Fig. 5b), the SNR in the former case should be better than that in the latter case. Such an enhanced SNR will directly lead to improvement of the measurement precision in ellipsometry. Unfortunately, the limited dynamic range of the PD obscures these effects in the present results. However, if the optical comb spectrum were to be optimized to this limited dynamic range by means of spectral shaping[38] or mode filtering[39], the SNR enhancement caused by the noise rejection would be clearer.

We next discuss the measurement time for an ellipsometric evaluation in DCSE. The measurement time depends on the frequency difference between the repetition rates of the two comb lasers ($\Delta f_{rep}$) and the number of times signal averaging is performed on the interferograms. Since the interferograms in DCSE are observed every $1/\Delta f_{rep}$, the measurement time with signal averaging is equal to the product of $1/\Delta f_{rep}$ and the number of signal averaging operations. In our demonstration, we employed a 21 Hz frequency difference between the repetition rates of the two comb lasers and 1000 signal averaging operations to achieve a sufficient SNR. The measurement time for ellipsometric evaluation was approximately 48 s, including the time for polarization analysis using the amplitude and phase of two orthogonal axes and spectral decoding by means of Fourier transformation with a high spectral resolution of $1.2 \times 10^{-5}$ nm (1.5 MHz). At this $\Delta f_{rep}$, we can extend the observable spectral bandwidth up to 55 THz because of the large frequency-scale magnification ($= f_{rep}/\Delta f_{rep}$), although in our demonstration, the spectral bandwidth was limited to 5–10 THz (40–80 nm) to ensure sufficient intensity of the comb laser sources. Since there is a trade-off relationship between the frequency difference between the repetition rates of the two comb lasers and the observable spectral bandwidth because of the sampling theorem related to the observation of interferograms using the dual-comb spectroscopic scheme, this trade-off can be optimized in terms of the spectral bandwidth for DCSE in the same way as for dual-comb spectroscopy[29]. The optimized frequency difference between the repetition rates is estimated to be 230 Hz or 115 Hz

for a spectral bandwidth of 5 THz (40 nm) or 10 THz (80 nm), respectively. These results indicate that DCSE can be performed 5–10 times faster than our demonstration. Furthermore, the required number of signal averaging operations was determined to ensure a sufficient SNR of the interferograms, which depends on the experimental conditions, such as the optical throughput of the system, the stability of the optical comb lasers, and the characteristics of the detectors and spectra such as the spectral resolution and bandwidth. This indicates that faster DCSE evaluation could be achieved by means of single-shot measurements of the interferograms. Although the minimum measurement time for DCSE depends on the experimental conditions, fast ellipsometric evaluation using DCSE with high spectral resolution and accuracy and a wide spectral bandwidth can be by virtue of the automatic time-sweep nature of DCSE for both polarization analysis and spectral decoding.

Finally, we discuss the uneven spectral shape in the present DCSE system, as shown in Fig. 1b. Some lower-intensity regions were observed in the optical frequency, which decreased the SNR of the ellipsometric parameters. We eliminated these wavelength regions using the criterion of an SNR of 10 for the s-polarization spectra. Although we used polarization optics to impose a linear polarization on the laser output, the laser sources showed a wavelength dependency of the polarization state, namely, polarization dispersion, because of wavelength-dependent birefringence in the optical fiber. This resulted in the large fluctuation in the spectral shape shown in Fig. 1b. To enable the use of the entire spectrum, we could perform fine spectral and polarization shaping by means of a fine polarization controller set in front of the laser output, polarization-maintaining fiber-based laser sources, or laser sources containing a polarization beam splitter (PBS) as an output coupler. Such a modification of the laser sources would merely manipulate the output laser spectrum after the polarizer while not affecting the capability for ellipsometric evaluation. Other sources of polarization modification, such as the oblique incidence of the light on mirrors or other optics, could be compensated using a standard material because the polarization measurement in DCSE is based on the amplitude and phase spectra along two orthogonal axes. The correction of these polarization modifications would enable the full use of the high resolution, broadband amplitude, and phase spectra for a more precise ellipsometric evaluation.

In conclusion, we developed a polarization-modulation-free ellipsometric material evaluation method employing dual-comb spectroscopy with a spectral resolution of $1.2 \times 10^{-5}$ nm (1.5 MHz) over 1514–1595 nm (188–198 THz), and we presented a proof-of-principle demonstration of DCSE using birefringent materials and thin-film samples. An accuracy of 38.4 nm and a precision of 3.3 nm were achieved in the determination of the thicknesses of the thin-film samples. Although the accuracy and precision were somewhat worse than those of a commercially available SE system with a much broader spectral bandwidth, DCSE still has sufficient room for improvement of the accuracy and precision by means of broadband dual-comb sources[28]. Furthermore, the demonstration of DCSE in other optical regimes will expand its scope of application. An optical frequency comb can be coherently converted into other wavelength regions, including the ultraviolet, visible, infrared, and terahertz regions, by virtue of its generation mechanism of phase locking to a standard frequency, indicating that DCSE should also be applicable at wavelengths from the ultraviolet to the THz region. We expect that our method will be a powerful tool for materials science beyond the conventional limit of ellipsometry.

## Methods

**Experimental setup.** The optical setup of the DCSE system is shown in Fig. 1. Two highly stable home-made erbium-based mode-locked fiber lasers were employed as the signal and local comb oscillators, which have been described previously[28]. Each fiber laser consisted of an EOM, a piezoelectric transducer, and a Peltier thermo controller for the long-term broadband stabilization of the laser cavity. A delay line was also installed for the tuning of the repetition rate of the pulse train in the laser cavity. The repetition rates of the two combs were set to approximately 48 MHz, and the frequency difference of 21 Hz was precisely stabilized by means of phase locking between a well-stabilized 1.54 μm continuous wave (CW) laser and the nearest comb mode. The carrier-envelope offset frequency of each comb, which was detected using a $f$–$2f$ configuration[40], was phase-locked to a reference frequency by controlling the injection current of the pump lasers for the comb oscillator. Our stabilization system for the repetition rate and the carrier-envelope offset frequency achieved a narrow, sub-Hz relative linewidth between the combs.

The laser outputs of the individual comb lasers for the DCSE measurements were individually amplified by erbium-doped fiber amplifiers and then spectrally broadened with highly nonlinear fibers. The polarization states of the two optical comb lasers (the signal and local combs) were set to linear polarizations at 45° and −45°, respectively. The signal comb was incident on the sample in the reflection or transmission configuration, and was spatially overlapped with the local comb laser using a beam splitter. The interferograms of the p-polarization and s-polarization (or $x$-polarization and $y$-polarization) components of the comb lasers were individually detected using a PBS and InGaAs PDs. The interferograms were averaged 1000 times in the time domain using a coherent averaging condition[41]. Furthermore, we performed real-time phase compensation of the carrier phase drift of interferograms and the frequency drift of the CW laser as described previously[28]. The interferograms were processed via fast Fourier transformation using a computer to reconstruct the amplitude and phase spectra.

**Evaluation of the ellipsometric parameters via DCSE.** The polarization state of light propagating through our system can be represented in terms of the Jones calculus as:

$$\mathbf{E}_{obs} = SP_S(\theta_S)\mathbf{E}_S + P_L(\theta_L)\mathbf{E}_L,$$ (4)

where $\mathbf{E}$, $S$, and $P$ represent electric fields, the sample, and polarizers oriented at angle $\theta$. The subscripts obs, S, and L refer to the observed position, the signal comb arm, and the local comb arm.

Since the rotation angles of the polarizers in the signal comb and local comb arms were set to $\pi/4$ and $-\pi/4$, respectively, and the amplitudes of the electric fields after the polarizers were $\overline{E}_L$ and $\overline{E}_S$, Eq. (4) can be expressed as

$$\mathbf{E}_{obs} = \frac{1}{\sqrt{2}}\begin{bmatrix} \overline{E}_L + \overline{E}_S\exp(-i\Delta)\sin\Psi \\ \overline{E}_L\exp(-i\pi) + \overline{E}_S\cos\Psi \end{bmatrix},$$ (5)

where

$$S = \begin{bmatrix} \exp(-i\Delta)\sin\Psi & 0 \\ 0 & \cos\Psi \end{bmatrix}.$$ (6)

We observed the interferograms of each polarization component by using a PBS and extracting the cross-correlation term between the signal and local combs in Eq. (5). Since Fourier transformation of the interferograms decodes the amplitude and phase of the cross-correlation term, $\Psi$ and $\Delta$ are directly calculated along with the amplitude and phase spectra, as shown in Eqs. (4), (5). We eliminated certain wavelength regions for the ellipsometric evaluation based on the low amplitude of the s-polarization spectrum. The elimination criterion was an SNR of 10 or less, where the SNR was defined as the spectral amplitude divided by the standard deviation of the noise spectral region.

**Samples.** The fast axis of a Soleil-Babinet compensator (SBC-IR, Thorlabs, Inc.) was aligned at 45° along the $x$-axis of the sample coordinate system. To obtain a theoretical estimate of the retardation of the Soleil-Babinet compensator, relative wedge distances of 0–25 mm of the Soleil-Babinet compensator were calibrated at 1550 nm (193.4 THz) using the phase obtained via DCSE. DCSE evaluation of the Soleil-Babinet compensator was performed in 2.5 mm steps over relative wedge distances of up to 25 mm. For the theoretical estimation of the retardation of the Soleil-Babinet compensator, the ordinary and extraordinary refractive indices of the Soleil-Babinet compensator were estimated using a Sellmeier model with the coefficients provided by the manufacturer (Thorlabs, Inc.).

A custom-made high-order wave plate was designed as a quarter-wave plate at 633 nm, which also functioned as a high-order wave plate at 1550 nm (193.4 THz), as demonstrated in this study. DCSE evaluation of the high-order wave plate was performed in 10° rotation angle increments over 360°.

The thicknesses of two film samples (STA-6000-C and STA-9000-C, Five Lab Co., Ltd.) were measured in the reflection configuration. The incidence angle of the light was aligned at 58°, near the Brewster's angle of SiO₂ (55.2°), which was

confirmed by comparing the theoretical estimates and observed $\Psi$ and $\Delta$ values for thin-film samples with thicknesses of 0–900 nm. Film thicknesses of 0, 22, 59, 91, 109, 204, 405, 584, and 881 nm were used for the DCSE evaluation. The film thicknesses of these samples were determined beforehand using a commercially available single-wavelength ellipsometer (MARY102, Five Lab Co., Ltd., wavelength of 632.8 nm, incidence angle of 70°) calibrated using a NIST-traceable film thickness standard (Silicon Dioxide Film Thickness Standards, VLSI Standards, Inc.); the values thus obtained are plotted on the horizontal scale in Fig. 4c. We then performed film thickness determination using our DCSE system (spectral range of 1514–1595 nm, spectral resolution of $1.2 \times 10^{-5}$ nm, spectral spacing of $3.8 \times 10^{-4}$ nm, number of data points of approximately 200,000, incidence angle of 58°) and a commercially available SE system (M-2000DI-YK, J.A. Woollam, spectral range of 1000–1689 nm, spectral resolution of 3.4 nm coinciding with the spectral spacing, number of data points of 200, incidence angle of 58°) for comparison. The theoretical estimates were calculated using a three-layer model with a structure of air/SiO₂/Si, where the SiO₂ thickness ranged from 0 to 1000 nm and the thicknesses of the air and Si layers were set to infinity. The complex refractive indices of air, SiO₂, and Si were assumed to be 1.00, 1.46, and $3.87-i1.46\times10^{-2}$, respectively.

**Data availability.** The data that support the findings of this study are available from the corresponding author upon reasonable request.

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

## Acknowledgements

This work was supported by Exploratory Research for Advanced Technology (ERATO) MINOSHIMA Intelligent Optical Synthesizer Project (JPMJER1304) from Japan Science and Technology Agency (JST), Japan. The authors gratefully acknowledge Dr. Toshiyasu Tadokoro of Techno-Synergy, Inc. for his helpful comments and suggestions on this manuscript. The authors also appreciate the help from Mr. Takayuki Kimura of Yamaguchi University Nanofabrication Platform in Nanotechnology Platform Project sponsored by Ministry of Education, Culture, Sports, Science and Technology (MEXT), Japan, for the observation with the commercially available SE. The authors also thank Ms. Natsuko Takeichi and Ms. Shoko Lewis of Tokushima University for their help in the preparation of the manuscript. This article is dedicated to the late Dr. Atsushi Onae of National Institute of Advanced Industrial Science and Technology (AIST) to acknowledge his many scientific contributions to optical frequency metrology including optical comb.

## Author contributions

T.Y. and T.I. conceived the project. T.M., Y.-D.H., T.I., and T.Y. designed the experiments. T.M., Y.-D.H., K.S., E.H., Y.K., and S.O. performed the experiments. S.O. and H.I. constructed the dual-comb system. T.M. and Y.-D.H. analyzed the data. T.M. wrote the manuscript. Y.M., H.Y., T.I., and T.Y. contributed to manuscript preparation. All authors discussed the results and commented on the manuscript.

## Additional information

**Competing interests:** The authors declare no competing financial interests.

