## [Peer Review File · Nature Communications]

Reviewers' comments:

Reviewer #1 (Remarks to the Author):

The work by Minamikawa and coworkers reports on the extension of the Dual-comb spectroscopy to the specific case of ellipsometry, which is an interesting and an emerging new spectroscopic tool that exploits the frequency resolution, frequency accuracy, broad bandwidth, and brightness of frequency combs for ultrahigh-resolution, high-sensitivity broadband spectroscopies.

-The authors have presented already their results discussing the same examples in the paper "Dual-Optical-Comb Spectroscopic Ellipsometry" by Takeo Minamikawa, Yi-Da Hsieh, KYUKI SHIBUYA, Yoshiki Kaneohe, Sho Okubo, Hajime Inaba, Yasuhiro Mizutani, Takeshi Yasui, and Tetsuo Iwata

OSA Technical Digest (2016) (Optical Society of America, 2016), paper SW1H.5•https://doi.org/10.1364/CLEO_SI.2016.SW1H.5

The validity of their approach to the analysis of other examples should be presented and discussed to strengthen the paper.

-The use of the concept of "conventional ellipsometry" is misleading and confusing through the text, and the authors should be more specific about what they want to emphasize

-While the manuscript emphasizes the ultra-high spectra resolution and accuracy for deducing polarization states coming from absence of mechanical movements, aspects related to depolarization of the lasers light sources are not discussed.

-The manuscript emphasizes several times throughout the text the advantages of instrumental resolution, which however is not a novel concept and already well known from excellent reviews in dual-comb spectroscopy; but, the discussion is unclear of how this ultra-high resolution translates in a more accurate analysis of real samples. For example, in discussing the SiO₂ thin film standards, the discussion does not make clear if the uncertainty and errors on the SiO₂ thickness is lower than "conventional" spectroscopy.

-Also, despite the instrumental "ultra-high resolution", the analysis of birefringence and thickness requires the application of optical models, using dielectric functions not known with good accuracy in the IR and Terahertz range. Therefore, there is somehow a disconnection between the instrumental part/resolution and the analysis of data that is not well presented in the manuscript.

-As a form of communication, the formalism is presented in a way that does not attract a broad readership and it is not clear or accessible to nonspecialists in polarization based spectroscopies.

-As presented, the manuscript is of interest to readers limited only to the THz ellipsometry community, which is quite narrow at the moment, and since mainly emphasizes the instrumental part, it is more suitable for a specialized optics journal.

Therefore, I cannot recommend it for Nature Communication journal.

Reviewer #2 (Remarks to the Author):

The authors proposed a dual-comb spectroscopic ellipsometry, which broadened the application of dual-comb spectroscopy into a new field. This method may become an attractive topic for both communities of dual-comb spectroscopy and spectroscopic ellipsometry. In most applications of dual-comb spectroscopy, the investigators focused on amplitude spectrum but ignored phase spectrum. In comparison, the authors made full use of both spectra and applied in spectroscopic

ellipsometry. Consequently, all the advantages of dual-comb spectroscopy compared with conventional dispersive or Fourier transform spectroscopy are inherited to the present spectroscopic ellipsometry method. In other words, the present method can offer a much better spectral accuracy compared with conventional spectroscopic ellipsometry. To demonstrate the feasibility of this method, the authors measured three different samples. The results show the experimental setup works well as a spectroscopic ellipsometer. The manuscript is clearly written and the quality of presentation is good. The references give good insight into the mentioned subjects.

However, as mentioned by the authors, the important parameters of ellipsometry are Ψ and Δ . What is the substantial benefit when measuring Ψ and Δ (or analyzing the properties of samples) with a higher spectral accuracy? I cannot find the answer in the experimental results shown in this manuscript. It is suggested to have a comparison experiment with conventional spectroscopic ellipsometry. It would be interesting to show some results beyond the capacity of conventional methods.

Minor point:

In the discussion part, the authors mentioned the optimized frequency difference of the repetition rate is more than one hundred Hz, why did they use 21 Hz in the experimental system?

Reviewer #3 (Remarks to the Author):

The juvenile technique of dual comb spectroscopy has been used in an ellipsometric setup to characterize three different samples in their polarizing properties. I think, the idea behind the experiment is neat and the manuscript itself is well organized. Due to the uncommon usage of several technical terms and some mistakes (stated spectral resolution, typos, etc.), I would recommend the publication only after the authors will have eliminated those weaknesses.

They are as follows:

1. Title: The term "dual comb spectroscopy" is very common in the meantime, so that you can and even should change your "dual-optical-comb" phrase into "dual comb" in the title and throughout the whole manuscript. It will be more pleasant for the reader.
2. Abstract: In the end of your abstract, you state a spectral resolution of up to 1.2×10^{-5} nm across 5-10 THz. Please do not "jump" between the different units. Mention the spectral resolution in Hz (and in nm only in brackets if you like) as you do later in the manuscript. It is also common to state the relative resolution ($\Delta E/E$) or resolving power ($E/\Delta E$). And please pay attention: Your claimed 1.2×10^{-5} nm is not the "real" spectral resolution but the calculated value to which extend you can resolve the comb modes (I guess you calculated it by dividing the optical coverage by the measurement time and the repetition frequency). Your real spectral resolution is determined by your repetition frequency of 48 MHz. Please change the corresponding text passages accordingly.
3. Introduction: Second last sentence before equation 1: A d is missing at the end of determine: cannot be determined.
4. "Wavelength of each comb" does not make sense because it is broadband and you give a frequency at the end of the sentence. Do you mean "the frequency of each comb mode"?
5. Principle of operation: "had a periodic time delay interval of $\Delta T = 1/f_{repL} - 1/f_{repS}$. The equation is incorrect. It should be $\Delta T = 1/(f_{repL} - f_{repS})$. Please change it.
6. Sentence after equation 3: signal should be plural, signals.
7. End of this section: "spectral spacing of 48 MHz were achieved" this comes naturally since it is your repetition frequency. Please change the sentence to something like: "The comb modes are separated by the repetition frequency of 48 MHz and could be resolved down to 1.5 MHz (1.2×10^{-5})." Compare to my comment # 2.
8. Figure 1: Why is the spectral coverage of your lasers so narrow? You are saying that the lasers were presented elsewhere, in your citation [17]. There they covered 1000 to 1900 nm, which is 158 THz – 300 THz. Here it is "only" 189 THz – 194 THz and very modulated resulting in

unpleasant noise for example at ~ 189.8 THz, 191.2 THz and 192.7 THz. Can you comment on that in the manuscript, please?

9. Basic spectral performance: "with a one complete period of repetition rate" what is meant by that? Maybe it will get clearer when you cite the value of this period.

10. Please change "ultra-high spectral resolution" to "resolved comb lines" and "spectral spacing of 48 MHz", according to comment # 2.

11. Soleil-Babinet compensator: Figure 2 The vertical axis for the ellipsometric parameters are unfavorably chosen. It is not possible to see the detailed changes vs. optical frequency. The movement in the videos are too fast (half or one third of the speed would be better to have time to study the details), and also here the vertical axis should be different (Psi axis going from 0.5 rad to 1.5 rad). Maybe of minor importance: can you change the background from black to white and the font into black in the manuscript figures? This would be more economical for printouts...

12. Figure 3: are the uncovered slices due to the spectrum modulations? Please comment how you chose to cut the spectrum. What was your signal to noise criteria? What is your SNR for the three measurements in general? Please include it to all three sample sections.

13. Figure 4: Also here the vertical axes should be chosen to see more details (zoomed in), maybe different perspective. Can you change also here the background to white? This would match better to panel c. Why are the errors so different in panel c? Can you comment on that?

14. Discussion: Amplitude and phase could be determined with a spectral resolution of 48 MHz and not 1.5 MHz, see again comment #2.

15. Methods, first line: can you add if it were commercial or self built lasers? Here you could mention why the lasers do not cover 1000 to 1900 nm.

Responses to reviewers:

Reviewer #1:

Thank you very much for your helpful comments regarding our study. We have attempted to respond to each of your comments:

- *The authors have presented already their results discussing the same examples in the paper “Dual-Optical-Comb Spectroscopic Ellipsometry” by Takeo Minamikawa, Yi-Da Hsieh, Kyuki Shibuya, Yoshiki Kaneohe, Sho Okubo, Hajime Inaba, Yasuhiro Mizutani, Takeshi Yasui, and Tetsuo Iwata, OSA Technical Digest (2016) (Optical Society of America, 2016), paper SW1H.5•https://doi.org/10.1364/CLEO_SI.2016.SW1H.5, The validity of their approach to the analysis of other examples should be presented and discussed to strengthen the paper.*

In the conference proceeding you mentioned, we only presented a possibility of the spectroscopic ellipsometry using dual-optical-comb spectroscopy. We utilized the limited optical frequency for the ellipsometric evaluation, and the ellipsometric evaluation of each optical frequency was performed in the same manner of the single-wavelength ellipsometry. Furthermore, the quantitative ellipsometric evaluation capability was not given, and only the spectral resolution was discussed as the same manner of general dual-optical-comb spectroscopy.

In this manuscript we submitted to *Nature Communications*, we extended to the ellipsometric evaluation utilizing whole spectra with high spectral resolution. Furthermore, we also provided further results and discussions of the ellipsometric evaluations with dual-optical-comb spectroscopy, in which the main results and conclusion are not apparent from the conference proceeding.

We also think that we can follow the duplicate publication policy of Nature journal:

Publication ethics > Duplicate publication

Nature journals allow publication of meeting abstracts before the full contribution is submitted. Such abstracts should be included with the Nature journal submission and referred to in the cover letter accompanying the manuscript. This policy does not extend to meeting abstracts and reports available to the media or which are otherwise publicised outside the scientific community during the submission and consideration process.

But anyway, we have to refer to it in the cover letter and include the proceeding. In the revised submission, we attached the proceeding with the cover letter.

- *The use of the concept of “conventional ellipsometry” is misleading and confusing through the text, and the authors should be more specific about what they want to emphasize*

We defined the multi-channel dispersive spectrometer equipped with polarization modulation as the conventional spectroscopic ellipsometry.

The mechanical polarization modulation limits the mechanical stability, thermal stability, and spectral bandwidth of simultaneous spectroscopic observation in ellipsometric evaluation. Although the electro-optical polarization modulation may be used for non-mechanical measurement, wavelength dependency of polarization rotation is another problem. We would like to emphasize that our proposed method can overcome the limitation in spectral resolution/accuracy and requirement in the polarization modulation by use of the full mode-resolved spectrum of amplitude and phase with dual-comb spectroscopy.

To avoid the misleading of the term of the conventional ellipsometry, the Introduction section was totally modified, i.e., we predominantly focused on spectroscopic ellipsometry in the Introduction section, and described the brief principles and limitations of the conventional spectroscopic ellipsometry. We also defined the term of “conventional ellipsometry” used in this manuscript on line 15 in page 5. The term of “conventional spectroscopic ellipsometry” is now only used to refer to the conventional spectroscopic ellipsometry defined above.

- *While the manuscript emphasizes the ultra-high spectra resolution and accuracy for deducing polarization states coming from absence of mechanical movements, aspects related to depolarization of the lasers lights sources are not discussed.*

To avoid the depolarization effect of the laser source itself, we used polarizers at the laser output. The depolarization of the laser sources was effected only for the spectral shape after the polarizers. In this study, we found lower spectral intensity region that resulted the decrease of the SNR of ellipsometric parameters. This unwanted effect can be avoided by the fine spectral shaping with a fine polarization controller, the polarization maintained fiber-based laser sources, or the laser sources containing a polarization beam splitter as an output coupler. This modification of the laser sources offers only for the manipulation of output laser spectrum after the polarizer, but does not effected the capability of the ellipsometric evaluation.

After the polarizer, if the polarization state of light is changed due to the oblique incidence of light to mirrors and other optics, we can compensate its polarization state with a standard material because of the polarization measurement based on the amplitude and phase spectra along two orthogonal axes in our proposed method.

We appended the discussion of the effect of the depolarization of laser sources in Discussion section on line 3 in page 19.

We also appended the sentence “We eliminated the wavelength region for the ellipsometric evaluation due to the low amplitude of the *s*-polarization spectrum. The elimination criterion was the SNR of 10 or less that defined as the spectral amplitude divided by the standard deviation of noise spectral region.” on line 14 in page 23.

- *The manuscript emphasizes several times throughout the text the advantages of instrumental resolution, which however is not a novel concept and already well known from excellent reviews in dual-comb spectroscopy; but, the discussion is unclear of how this ultra-high resolution translates in a more accurate analysis of real samples. For example, in discussing the SiO₂ thin film standards, the discussion does not make clear if the uncertainty and errors on the SiO₂ thickness is lower than “conventional” spectroscopy.*

According to your comment, we discussed again for the capability of our proposed method. One important and novel concept of this study is the simultaneous realization of ellipsometric evaluation with wide spectral bandwidth, high-spectral resolution, and no polarization modulation. High-spectral resolution of the dual-comb spectroscopy was described in the several reviews as you mentioned; however, the capability of the polarization-modulation-free ellipsometric evaluation with wide spectral bandwidth and high spectral resolution was not mentioned in the previous study. This can be only achieved by making full use of mode-resolved optical comb spectra of amplitude and phase with dual-comb spectroscopy.

The importance of our demonstration, especially in the SiO₂ thin film standards, is the ellipsometric evaluation without polarization modulation. Although the errors on the SiO₂ thickness was lower than the conventional spectroscopic ellipsometry, the polarization-modulation-free ellipsometric evaluation of its thickness was not realized so far. Furthermore, we also found that the ellipsometric evaluation with high spectral resolution enables the enhancement of signal-to-noise ratio owing to the noise rejection effect of discretely localized comb modes. As shown in Fig. 5 in the revised manuscript, the noise components existing in the frequency gap between comb modes can be rejected in a mode-resolved optical comb spectrum, while is still included in a broadband continuous spectrum with low spectral resolution in the conventional spectroscopic ellipsometry.

This discussion implies that the ellipsometric evaluation using the DCSE with high spectral resolution will directly link to the improvement of the measurement precision and accuracy in the ellipsometry.

To emphasize the capability of our proposed method, *i.e.* wide-spectral range, high-spectral resolution, and no polarization modulation, we appended the words of “wide spectral bandwidth, ultra-high spectral resolution, and no polarization modulation” or the like to explain the capability of the proposed method throughout the manuscript. Furthermore, we also appended the discussion about the SNR enhancement effect with a comb-resolved spectrum in Discussion section on line 11 in page 15.

Fig. 5 SNR enhancement in ellipsometric measurement with high spectral resolution. (a) Rejection of noise component existing in the frequency gap between comb modes is applicable in a mode-resolved spectrum. (b) No noise rejection is applicable with low spectral resolution.

- Also, despite the instrumental “ultra-high resolution”, the analysis of birefringence and thickness requires the application of optical models, using dielectric functions not known with good accuracy in the IR and Terahertz range. Therefore, there is somehow a disconnection between the instrumental part/resolution and the analysis of data that is not well presented in the manuscript.

One of the benefit of the ultra-high spectral resolution is the SNR enhancement effect as discussed in the reply on your comment #4. The other is the ellipsometric evaluation of fine spectral response of materials such in IR and Terahertz range. As you mentioned, in IR and Terahertz range, dielectric functions are not known with good accuracy for now, so we should consider several models, such as Lorentz model, Debye model, Drude model, and others, for the ellipsometric evaluation of materials. After the theoretical models are established, our proposed method must be a powerful method for the ellipsometric evaluation in IR and Terahertz range. Furthermore, the ellipsometric parameters of Ψ and Δ can be obtained without the theoretical model of dielectric functions, and so the actual spectral dependency of the ellipsometric parameters obtained with the proposed method might be also helpful for the construction of theoretical models of dielectric functions owing to the high spectral resolution capability.

To show the capability of our proposed method for the unknown or ambiguous theoretical models of dielectric functions, we appended the sentence, “Even if the theoretical models of dielectric functions are not well known, the DCSE can evaluate the spectral dependency of the ellipsometric parameters of Ψ , Δ at least. This spectral dependency of the ellipsometric parameters with high spectral resolution might be a helpful data for the estimation of theoretical models.” in Discussion section on line 2 in page 15.

- *As a form of communication, the formalism is presented in a way that does not attract a broad readership and it is not clear or accessible to nonspecialists in polarization based spectroscopies.*

For the easy understanding of the importance and usefulness of the ellipsometry for non-specialists in polarization based spectroscopies, Abstract was modified, and Introduction section was reformed to include the typical applications of ellipsometry on line 4 in page 5.

Furthermore, both the wavelength (nm) and the optical frequency (THz) were wrote to explain the spectral resolution, spectral bandwidth, and the others.

- *As presented, the manuscript is of interest interested to readers limited only to the THz ellipsometry community, which is quite narrow at the moment, and since mainly emphasize the instrumental part, it is more suitable for a specialized optics journal.*

Our proposed DCSE scheme can be applicable to all the wavelength region including ultraviolet, visible, near-infrared, infrared, and terahertz regions in terms of instrumental methodology, even if we provided a proof-of-principle demonstration of the DCSE in only near-infrared region (around 1550 nm) in this study. Importantly, optical frequency comb in these wavelength regions can be coherently linked because these are referenced to the frequency standard of an atomic clock. Furthermore, the applications of ellipsometry include the wide variety of academic and industrial fields of such as thin film science, semiconductor, biology, graphenics, and more, indicating there are a lot of potential users of the DCSE.

Our findings of the capability of the DCSE in this study will serve as a basis for accelerating the academic and industrial research by means of ellipsometry, and so we believe that our results have general interest for the wide variety of community.

To get interested in our study for wide variety of readers, we appended the explanation of the typical applications of ellipsometry on line 4 in page 5.

To show the applicability to other wavelength region from ultraviolet to terahertz region, we also appended the sentence “The optical-frequency-comb can be coherently converted to the other wavelength region including ultraviolet, visible, infrared, and terahertz regions owing to its generating mechanism of phase-locking to a frequency standard, indicating the DCSE is also applicable to the wavelength region from ultraviolet to THz region.” on line 6 in page 20.

Reviewer #2 (Remarks to the Author):

Thank you very much for your helpful comments regarding our study. We have attempted to respond to each of your comments:

- However, as mentioned by the authors, the important parameters of ellipsometry are Ψ and Δ . What is the substantial benefit when measuring Ψ and Δ (or analyzing the properties of samples) with a higher spectral accuracy? I cannot find the answer in the experimental results shown in this manuscript. It is suggested to have a comparison experiment with conventional spectroscopic ellipsometry. It would be interesting to show some results beyond the capacity of conventional methods.

According to your comment, we discussed again for the capability of our proposed method. One important and novel concept of this study is the simultaneous realization of ellipsometric evaluation with wide spectral bandwidth, high-spectral resolution, and no polarization modulation. High-spectral resolution of the dual-comb spectroscopy was described in the several reviews as you mentioned; however, the capability of the polarization-modulation-free ellipsometric evaluation with wide spectral bandwidth and high spectral resolution was not mentioned in the previous study. This can be only achieved by making full use of mode-resolved optical comb spectra of amplitude and phase with dual-comb spectroscopy.

Furthermore, we also found that the ellipsometric evaluation with high spectral resolution enables the enhancement of signal-to-noise ratio owing to the noise rejection effect of discretely localized comb modes. As shown in Fig. 5 in the revised manuscript, the noise components existing in the frequency gap between comb modes can be rejected in a mode-resolved optical comb spectrum, while is still included in a broadband continuous spectrum with low spectral resolution in the conventional spectroscopic ellipsometry.

This discussion implies that the ellipsometric evaluation using the DCSE with high spectral resolution will directly link to the improvement of the measurement precision and accuracy in the ellipsometry.

To emphasize the capability of our proposed method, *i.e.* wide-spectral range, high-spectral resolution, and no polarization modulation, we appended the words of “wide spectral bandwidth, ultra-high spectral resolution, and no polarization modulation” or the like to explain the capability of the proposed method throughout the manuscript. Furthermore, we also appended

the discussion about the SNR enhancement effect with a comb-resolved spectrum in Discussion section on line 11 in page 15.

We also compared the results of the SiO₂ thickness estimated by the DCSE and a commercially available ellipsometer as shown in the revised Fig. 4c. This result indicated that the DCSE can estimate the film thickness as similar to the general spectroscopic ellipsometry. Importantly, the DCSE estimated the film thickness without any polarization modulation, while the general spectroscopic ellipsometry required mechanical or electro-optic polarization modulation to analyse polarization state.

We appended the sentence “The film thicknesses of the thin film standards were determined by a commercially available single-wavelength ellipsometer (MARY102, Five Lab, Co., Ltd.), and was compared with the estimates by the DCSE in this study.” on line 8 in page 25. According to this result, we changed the thickness values of the film standards to the estimated value by the commercially available single-wavelength ellipsometer.

We also appended the sentence “Although the error correction should be performed for practical use of the DCSE, this result indicated that the DCSE can estimate the film thickness as similar to the general spectroscopic ellipsometry. Importantly, the DCSE estimated the film thickness without any polarization modulation, while the general spectroscopic ellipsometry required mechanical or electro-optic polarization modulation to analyse polarization state.” on line 12 in page 13.

Fig. 5 SNR enhancement in ellipsometric measurement with high spectral resolution. (a) Rejection of noise component existing in the frequency gap between comb modes is applicable in a mode-resolved spectrum. (b) No noise rejection is applicable with low spectral resolution.

Minor point:

- *In the discussion part, the authors mentioned the optimized frequency difference of the repetition rate is more than one hundred Hz, why did they use 21 Hz in the experimental system?*

For the DCSE evaluation with wider spectral bandwidth, lower frequency difference of the repetition rate is required. In the lower frequency difference of the repetition rate, the interferogram tends to be affected by timing jitter due to the coherence time between the two comb lasers. To demonstrate the capability of the DCSE evaluation with lower frequency difference of the repetition rate, we used the frequency difference of 21 Hz in this study. The higher frequency difference of the repetition rate can reduce the unwanted effect by the timing jitter, whereas the spectral bandwidth is limited. The frequency difference of the repetition rate and the spectral bandwidth thus have a trade-off relationship because of the sampling theorem in the observation of interferograms with the dual-comb spectroscopic scheme, indicating the optimum frequency difference of the repetition rate must exist depending on the spectral bandwidth.

We appended the sentence “At this Δf_{rep} , we can extend the observable spectral bandwidth up to 55 THz due to the large frequency-scale magnification ($=f_{\text{rep}}/\Delta f_{\text{rep}}$), although our demonstration limited its spectral bandwidth up to 5 to 10 THz (40 to 80 nm) owing to obtain sufficient intensity of the comb laser sources.” on line 14 in page 17. We also changed the sentence to “Since the frequency difference of the repetition rate of the two comb lasers and the observable spectral bandwidth has a trade-off relationship because of the sampling theorem in the observation of interferograms with the dual-comb spectroscopic scheme, that can be optimized in terms of spectral bandwidth for DCSE in the same way as dual-comb spectroscopy.” on line 2 in page 18.

Reviewer #3:

Thank you very much for your helpful comments regarding our study. We have attempted to respond to each of your comments:

- Title: The term “dual comb spectroscopy” is very common in the meantime, so that you can and even should change your “dual-optical-comb” phrase into “dual comb” in the title and throughout the whole manuscript. It will be more pleasant for the reader.

As you mentioned, we changed the title to “Dual-comb spectroscopic ellipsometry”. We also changed to “dual-comb spectroscopy” in the manuscript.

- Abstract: In the end of your abstract, you state a spectral resolution of up to 1.2×10^{-5} nm across 5-10 THz. Please do not “jump” between the different units. Mention the spectral resolution in Hz (and in nm only in brackets if you like) as you do later in the manuscript. It is also common to state the relative resolution ($\Delta E/E$) or resolving power ($E/\Delta E$). And please pay attention: Your claimed 1.2×10^{-5} nm is not the “real” spectral resolution but the calculated value to which extend you can resolve the comb modes (I guess you calculated it by dividing the optical coverage by the measurement time and the repetition frequency). Your real spectral resolution is determined by your repetition frequency of 48 MHz. Please change the corresponding text passages accordingly.

In this study, we defined the spectral resolution as the linewidth of the comb modes as the same way of Ref. 32. It is because the ultimate resolution of dual-comb spectroscopy can be reached to the linewidth, while the spectral sampling point is discrete with the spectral spacing that is coincided with the repetition rate of comb sources.

To avoid the confusion of the definition of “spectral resolution”, we appended the sentence “Here, we define the spectral resolution as the linewidth of the comb modes³³. It is because the ultimate resolution of dual-comb spectroscopy can be reached to the linewidth, while the spectral sampling point is discrete with the spectral spacing that is coincided with the repetition rate of comb sources.” on line 1 in page 9, and also the reference #32.

[32] Hebert NB, Boudreau S, Genest J, Deschenes JD. Coherent dual-comb interferometry with quasi-integer-ratio repetition rates. *Opt. Express* 2014, **22**(23): 29152-29160.

- Introduction: Second last sentence before equation 1: A d is missing at the end of determine: cannot be determined.

Thank you for your editing of our manuscript. Actually, Introduction section was totally modified, and so this word was deleted for now. The other words were also checked and corrected to proper words.

- “Wavelength of each comb” does not make sense because it is broadband and you give a frequency at the end of the sentence. Do you mean “the frequency of each comb mode”?

As you mentioned, we meant “the frequency of each comb mode”, so we change the sentence to “the frequency of each comb mode”.

- Principle of operation: “had a periodic time delay interval of $\Delta T = 1/f_{repL} - 1/f_{repS}$. The equation is incorrect. It should be $\Delta T = 1/(f_{repL} - f_{repS})$. Please change it.

Actually, ΔT in the original manuscript means the sampling interval of interferogram, in other word, the time interval of sampling point by sampling point. To avoid the confusion, we changed to “a sampling interval”. Furthermore, we also changed to $\Delta t (=1/f_{rep,L} - 1/f_{rep,S})$, because ΔT is sometimes used for the interval of interferogram ($1/(f_{rep,S} - f_{rep,L})$).

- Sentence after equation 3: signal should be plural, signals.

Thank you for your editing of our manuscript. We have corrected the word of “signal” to the plural form of “signals” on line 6 in page 8.

- End of this section: “spectral spacing of 48 MHz were achieved” this comes naturally since it is your repetition frequency. Please change the sentence to something like: “The comb modes are separated by the repetition frequency of 48 MHz and could be resolved down to 1.5 MHz (1.2×10^{-5} nm).” Compare to my comment # 2.

As you mentioned, we changed the sentence of “...spectral spacing of 48 MHz were achieved” to “The comb modes are separated by the repetition frequency of 48 MHz (3.8×10^{-4} nm) and could be resolved down to 1.5 MHz (1.2×10^{-5} nm).” in the figure legend of Figure 1.

- Figure 1: Why is the spectral coverage of your lasers so narrow? You are saying that the lasers were presented elsewhere, in your citation [17]. There they covered 1000 to 1900 nm, which is 158 THz – 300 THz. Here it is “only” 189 THz – 194 THz and very modulated resulting in unpleasant noise for example at ~ 189.8 THz, 191.2 THz and 192.7 THz. Can you comment on that in the manuscript, please?

This is a proof-of-principle demonstration for the necessary first step of our proposal. So, we demonstrated the DCSE with the spectral range over 5-10 THz (40-80 nm in wavelength) to obtain sufficient signal-to-noise ratio. We thought that this spectral range is enough to demonstrate the capability of the proposed ellipsometric evaluation method with wide spectral bandwidth and high spectral resolution without mechanical movements.

We appended the sentence “At this Δf_{rep} , we can extend the observable spectral bandwidth up to 55 THz due to the large frequency-scale magnification ($=f_{\text{rep}}/\Delta f_{\text{rep}}$), although our

demonstration limited its spectral bandwidth up to 5 to 10 THz (40 to 80 nm) owing to obtain sufficient intensity of the comb laser sources.” on line 14 in page 17.

The noise observed at the region comes from the low intensity of the comb sources at this region. Actually, the comb sources were not optimized in terms of the spectral dispersion of polarization, leading a part of the spectral component of the comb sources was attenuated at the polarizers that employed in front of the laser output. This unwanted effect can be avoided by the fine spectral shaping with a fine polarization controller, the polarization maintained fiber-based laser sources, or the laser sources containing a polarization beam splitter as an output coupler. This modification of the laser sources offers only for the manipulation of output laser spectrum after the polarizer, but is not effected the capability of the ellipsometric evaluation.

We appended the discussion of the effect of the low intensity region in Discussion section on line 3 in page 19.

- Basic spectral performance: “with a one complete period of repetition rate” what is meant by that? Maybe it will get clearer when you cite the value of this period.

The one complete period of repetition rate meant $1/\Delta f_{\text{rep}}$, which is the one period of an interferogram.

We appended the term of “ $(1/\Delta f_{\text{rep}})$ ” on line 7 in page 9.

- Please change “ultra-high spectral resolution” to “resolved comb lines” and “spectral spacing of 48 MHz”, according to comment # 2.

Please see our reply on your comment #2.

- Soleil-Babinet compensator: Figure2 The vertical axis for the ellipsometric parameters are unfavorably chosen. It is not possible to see the detailed changes vs. optical frequency. The movement in the videos are too fast (half or one third of the speed would be better to have time to study the details), and also here the vertical axis should be different (Psi axis going from 0.5 rad to 1.5 rad). Maybe of minor importance: can you change the background from black to white and the font into black in the manuscript figures? This would be more economical for printouts...

As you mentioned, we changed to the proper vertical axis of Figure 2 and Movies 1 and 2. The background of figure and movies were also changed to white. We also changed the frame rate of the movies to one third as slower as that of the original movie.

As the same manner, the background colour, the vertical axis, and the frame rate of Figures 3 and 4, and Movies 3 to 6 were also changed.

- Figure 3: are the uncovered slices due to the spectrum modulations? Please comment how you chose to cut the spectrum. What was your signal to noise criteria? What is your SNR for the three measurements in general? Please include it to all three sample sections.

In this study, we found lower spectral intensity that resulted the increase of the noise of ellipsometric parameters. We eliminated these wavelength regions with the criterion of the signal-to-noise ratio of 10 or less, which defined as the spectral amplitude divided by the standard deviation of noise spectral region.

The typical signal-to-noise ratios in the demonstrations of Soleil-Babinet compensator, high-order waveplates, and thin film standards were 235, 800, and 246, respectively.

We appended the sentence “We eliminated the wavelength region for the ellipsometric evaluation due to the low amplitude of the *s*-polarization spectrum. The elimination criterion was the SNR of 10 or less that defined as the spectral amplitude divided by the standard deviation of noise spectral region.” on line 14 in page 23.

We also appended the typical signal-to-noise ratios of each sample in three sample sections.

- Figure 4: Also here the vertical axes should be chosen to see more details (zoomed in), maybe different perspective. Can you change also here the background to white? This would match better to panel c.

As you mentioned, we changed to the proper vertical axis of Figure 4 and Movies 5 and 6. The background of figure and movies were also changed to white. We also changed the frame rate of the videos to one third as slower as that of the original movie.

- Why are the errors so different in panel c? Can you comment on that?

We revised Fig. 4c to compare the results of the SiO₂ thickness estimated by the DCSE and a commercially available ellipsometer. According to the uncertainty of the calibration standards used for the commercially available ellipsometry, the uncertainty of the SiO₂ thickness was 2.1 nm at the thickness of 881 nm of the SiO₂ thin film. This uncertainty was much smaller than the errors observing with the DCSE. We think that the errors of the DCSE evaluation might be caused by the low signal-to-noise ratio of the ellipsometric parameters or the accuracy of the incident angle to the film standards. Although the error correction should be performed for practical use of the DCSE, this result indicated that the DCSE can estimate the film thickness as similar to the general spectroscopic ellipsometry. Importantly, the DCSE estimated the film thickness without any polarization modulation, while the general spectroscopic ellipsometry required mechanical or electro-optic polarization modulation to analyse polarization state.

We appended the sentence “Although the error correction should be performed for practical use of the DCSE, this result indicated that the DCSE can estimate the film thickness as similar to the general spectroscopic ellipsometry. Importantly, the DCSE estimated the film thickness without any polarization modulation, while the general spectroscopic ellipsometry required mechanical or electro-optic polarization modulation to analyse polarization state.” on line 12 in page 13.

- Discussion: Amplitude and phase could be determined with a spectral resolution of 48 MHz and not 1.5 MHz, see again comment #2.

Please see our reply on your comment #2.

- Methods, first line: can you add if it were commercial or self built lasers? Here you could mention why the lasers do not cover 1000 to 1900 nm.

The comb sources were self-built lasers, but we optimized the spectral coverage over 5-10 THz in this work to obtain sufficient signal-to-noise ratio. For the DCSE evaluation with wider spectral bandwidth, lower frequency difference of the repetition rate is required. To demonstrate the capability of the DCSE evaluation with lower frequency difference of the repetition rate, we used the frequency differential of 21 Hz in this study.

We appended the word of “home-made” on line 2 in page 21. We also appended the sentence “At this Δf_{rep} , we can extend the observable spectral bandwidth up to 55 THz due to the large frequency-scale magnification ($=f_{\text{rep}}/\Delta f_{\text{rep}}$), although our demonstration limited its spectral bandwidth up to 5 to 10 THz (40 to 80 nm) owing to obtain sufficient intensity of the comb laser sources.” on line 14 in page 17.

Reviewers' comments:

Reviewer #1 (Remarks to the Author):

As the authors state the "important and novel concept of this study is the simultaneous realization of ellipsometric evaluation with wide spectral bandwidth, high-spectral resolution, and no polarization modulation."

Although the authors have improved significantly the manuscript, still the limiting issue of this manuscript is that it does not clearly demonstrate through the examples the benefit of measuring Ψ and Δ with the higher spectral accuracy.

The direct evidence through a clear example of how the wide spectral bandwidth and high-spectral resolution translates in more accurate results is still missing.

Specifically, even the example claimed of SiO₂ in Fig. 4c, the authors claim the comparison with a commercial single-wavelength ellipsometer, although it is well known in the ellipsometry community the increase in accuracy when using spectroscopic ellipsometry. Therefore, the choice of the comparison is not the best possibility.

Additionally, it is still confusing in Fig. 4c captions (and related discussion) "Thickness prediction by DCSE". Is it thickness prediction ? or measurement? Please clarify. Also, it is unclear why some points have a vertical error bar and other a horizontal error bar.

Also, the authors mention the advantage of the absence of mechanical movement compared to conventional rotating elements-ellipsometry, but they do not enough consider/discuss comparison with accuracy of phase-modulated ellipsometry.

The authors to support their claims in further discussing some comments have mostly " appended the words of "wide spectral bandwidth, ultra-high spectral resolution, and no polarization modulation".

This is not enough.

In order to make validate and support their claim, the author should clearly show and compare in numbers the accuracy on Ψ and Δ and the resulting error on the refractive index for the birefringent material case and for the thickness of SiO₂ measured with both their approach and spectroscopic ellipsometry.

The English language should also be carefully revised, since some technical concepts are not grammatically well formulated, making them confusing.

Therefore, I cannot recommend publication of this manuscript in the present form.

Reviewer #2 (Remarks to the Author):

The authors addressed most of my concerns. The advantages of the present method are more convincing in the revised manuscript. But I do not agree the SNR enhancement shown in Fig.5 is due to high spectral resolution. Instead, it is resulted from the difference between discrete and continuous spectra.

Reviewer #3 (Remarks to the Author):

The authors Takeo Minamikawa et al. of the manuscript "Dual-comb spectroscopic ellipsometry" addressed all comments that had been given by the three referees. They provide a serious revision of the paper with modified explanations, figures and videos resulting in a much improved manuscript. I therefore recommend the publication of the manuscript in Nature Communications.

Responses to reviewers:

Reviewer #1:

Thank you very much for your helpful comments regarding our study. We have attempted to respond to each of your comments below:

- As the authors state the “important and novel concept of this study is the simultaneous realization of ellipsometric evaluation with wide spectral bandwidth, high-spectral resolution, and no polarization modulation.” Although the authors have improved significantly the manuscript, still the limiting issue of this manuscript is that it does not clearly demonstrate through the examples the benefit of measuring Ψ and Δ with the higher spectral accuracy. The direct evidence through a clear example of how the wide spectral bandwidth and high-spectral resolution translates in more accurate results is still missing. Specifically, even the example claimed of SiO_2 in Fig. 4c, the authors claim the comparison with a commercial single-wavelength ellipsometer, although it is well known in the ellipsometry community the increase in accuracy when using spectroscopic ellipsometry. Therefore, the choice of the comparison is not the best possibility.

We now compare our proposed method (DCSE, spectral range of 1514-1595 nm, spectral resolution of 1.2×10^{-5} nm, spectral spacing of 3.8×10^{-4} nm, number of data points of approximately 200,000) with the use of a commercially available spectroscopic ellipsometer (conventional SE, M-2000DI-YK, J.A. Woollam, spectral range of 1000-1689 nm, spectral resolution of 3.4 nm coinciding with the spectral spacing, number of data points of approximately 200) in terms of the root mean square error (RMSE) and precision of the repeated observation of the film thicknesses determined for the same set of thin-film samples, which correspond to the accuracy and precision, respectively, of the film thickness determination. The RMSE and thickness precision achieved via DCSE were 38.4 nm and 3.3 nm, respectively. By contrast, those achieved via conventional SE were 10.0 nm and 1.2 nm. Therefore, the RMSE and thickness precision achieved using the DCSE system were somewhat worse than those of the conventional SE system. However, it is important to note that there is a large difference in spectral bandwidth between the two systems considered in this comparison

and that the fitting accuracy for film thickness determination largely depends on the spectral range.

We also compared our proposed method with the use of the conventional SE with the same spectral range (spectral range of 1514-1595 nm, number of data points of 27). The resulting RMSE and thickness precision of the conventional SE were 28.9 nm and 12.1 nm, respectively. Therefore, RMSE achieved using DCSE is almost comparable to that using conventional SE whereas the thickness precision using DCSE was significantly better than that using conventional SE. A little difference of RMSE between them is mainly due to the fact that the instrumental calibration was not performed using the film thickness standard in DCSE. More precise calibration of the optical set-up of the DCSE system (incidence angle, systematic phase error, and so on) would reduce the systematic error and hence improve its performance.

In other words, DCSE has the potential to achieve moderate ellipsometric performance for a much narrower spectral bandwidth than that used in conventional SE systems. The present performance achieved in such a limited spectral range may benefit from the high spectral resolution of DCSE and/or the consequently higher number of spectral data points as well as the higher stability by virtue of the lack of polarization modulation. Since state-of-the-art dual-comb sources have a spectral bandwidth comparable to that available in conventional SE systems [28], the use of such broadband comb sources in DCSE is a promising way to enhance the performance to a level equal to or greater than that of conventional SE. Obviously, another important advantage of DCSE is the lack of need for any polarization modulation, which is required in conventional SE.

The results of the comparison between DCSE and conventional SE are shown in Fig. 4c and Supplementary Fig. 3. The details of the spectroscopic ellipsometer have been added to the Methods section. We have also modified the Results section (see line 7, page 14).

- *Additionally, it is still confusing in Fig. 4c captions (and related discussion) “Thickness prediction by DCSE”. Is it thickness prediction? or measurement? Please clarify.*

To avoid misleading the reader, we have changed “prediction” to “determination” in the captions of Fig. 4 and Supplementary Fig. 2 and in related sentences throughout the main manuscript.

- *Also, it is unclear why some points have a vertical error bar and other a horizontal error bar.*

We show only vertical error bars and no horizontal error bars. The values of the error bars, which represent the standard deviations, can be found in Supplementary Fig. 3b. The results indicate that our demonstration of film thickness determination involved some error in terms of measurement accuracy but that the measurement precision was good. A more detailed discussion of the accuracy and precision of the film thickness determination is given in our reply to comment #1.

- *Also, the authors mention the advantage of the absence of mechanical movement compared to conventional rotating elements-ellipsometry, but they do not enough consider/discuss comparison with accuracy of phase-modulated ellipsometry.*

As the reviewer mentioned, phase-modulated ellipsometry using a photoelastic modulator (PEM) or electro-optic modulator (EOM) offers the advantages of high precision, a short measurement time, and no mechanical rotation. However, when such phase-modulated ellipsometry is extended to spectroscopic ellipsometry, there are several drawbacks. First, since a general PEM or EOM shows a wavelength dependency in its phase modulation, precise calibration is required. The temperature dependency of the phase modulation is another problem. Second, the fast polarization modulation imposed by a PEM or EOM (typically, several tens of kHz to a few tens of MHz) makes it difficult to combine with the use of a multi-channel spectrometer equipped with a CCD or CMOS camera. The alternative combination of a fast photodetector with a monochromator to acquire the broad spectrum of the ellipsometric parameters hampers real-time spectroscopic ellipsometry. Therefore, DCSE has an advantage over phase-modulated ellipsometry in that it enables the broadband observation of the ellipsometric parameters in real time.

We have modified related sentences in the Introduction (see line 5, page 5). We have also added a comparison of the accuracies of DCSE and conventional ellipsometry, as described in our reply to comment #1. We were also initially lacking the term “photoelastic modulator” in the explanation of conventional non-mechanical ellipsometric measurements; we have added it on line 5, page 5.

The authors to support their claims in further discussing some comments have mostly “ appended the words of “wide spectral bandwidth, ultra-high spectral resolution, and no polarization modulation”. This is not enough. In order to make validate and support their claim, the author should clearly show and compare in numbers the accuracy on Ψ and Δ and the resulting error on the refractive index for the birefringent material case and for the thickness of SiO_2 measured with both their approach and spectroscopic ellipsometry.

We compare the RMSE and thickness precision between our DCSE system and a conventional SE system in place of the accuracy of Ψ and Δ for a more direct comparison of the thickness measurement performances of these spectroscopic ellipsometry techniques. See our reply to comment #1. We have also deleted "wide spectral bandwidth" from "wide spectral bandwidth, ultra-high spectral resolution, and no polarization modulation" because the spectral bandwidth in the present DCSE system is still smaller than that in conventional SE.

- The English language should also be carefully revised, since some technical concepts are not grammatically well formulated, making them confusing.

As you suggested, we have submitted the manuscript for proofreading and editing by a professional proofreading company. Please find the editing certificate attached.

Reviewer #2 (Remarks to the Author):

Thank you very much for your helpful comments regarding our study. We have attempted to respond to each of your comments below:

- *But I do not agree the SNR enhancement shown in Fig.5 is due to high spectral resolution. Instead, it is resulted from the difference between discrete and continuous spectra.*

As you noted, this SNR enhancement resulted from the difference between discrete and continuous spectra. To gain the benefit of the SNR enhancement provided by a discrete spectrum, an additional important factor is a high spectral resolution that is sufficient to realize spectral filtering of the comb mode, as described in the Discussion section and shown in Figure 5, even if the moderate spectral resolution that is achievable in typical spectrometers is sufficient for spectroscopic ellipsometry. Since the DCSE method makes it easy to achieve such a high spectral resolution by virtue of its automatic time-sweep nature, this SNR enhancement can be realized.

To emphasize the requirement of the high spectral resolution of DCSE for SNR enhancement, we have modified changed the relevant sentence to “The ultra-high spectral resolution of DCSE in acquiring the mode-resolved optical comb spectrum enables us to...” on line 3, page 18.

REVIEWERS' COMMENTS:

Reviewer #1 (Remarks to the Author):

The authors have properly addressed all commentss and implemeentations and significantly improved the manuscript.

Therefore, it is suitable for publication

Reviewer #2 (Remarks to the Author):

The authors addressed my concerns. It can be published as it is.